# Remaining Useful Life Prediction for Lithium-Ion Batteries Based on Improved Mode Decomposition and Time Series

Hairui Wang [1], Xin Ye [1], Yuanbo Li [1] and Guifu Zhu [2],*

1 Faculty of Information Engineering and Automation, Kunming University of Science and Technology, Kunming 650500, China; hrwang88@163.com (H.W.); yangzhubing_ye@kust.edu.cn (X.Y.); liyuanbo@kust.edu.cn (Y.L.)
2 Information Technology Construction Management Center, Kunming University of Science and Technology, Kunming 650500, China
* Correspondence: zhuguifu@kust.edu.cn

**Abstract:** Accurately predicting the remaining useful life (RUL) of lithium-ion batteries holds significant importance for their health management. Due to the capacity regeneration phenomenon and random interference during the operation of lithium-ion batteries, a single model may exhibit poor prediction accuracy and generalization performance under a single scale signal. This paper proposes a method for predicting the RUL of lithium-ion batteries. The method is based on the improved sparrow search algorithm (ISSA), which optimizes the variational mode decomposition (VMD) and long- and short-term time-series network (LSTNet). First, this study utilized the ISSA-optimized VMD method to decompose the capacity degradation sequence of lithium-ion batteries, acquiring global degradation trend components and local capacity recovery components, then the ISSA–LSTNet–Attention model and ISSA–LSTNet–Skip model were employed to predict the trend component and capacity recovery component, respectively. Finally, the prediction results of these different models were integrated to accurately estimate the RUL of lithium-ion batteries. The proposed model was tested on two public lithium-ion battery datasets; the results indicate a root mean square error (RMSE) under 2%, a mean absolute error (MAE) under 1.5%, and an absolute correlation coefficient ($R^2$) and Nash–Sutcliffe efficiency index (NSE) both above 92.9%, implying high prediction accuracy and superior performance compared to other models. Moreover, the model significantly reduces the complexity of the series.

**Keywords:** lithium-ion battery; variational mode decomposition; remaining useful life prediction; long and short-term time-series network

## 1. Introduction

Lithium-ion batteries have emerged as a primary power source in various industrial sectors, including mobile communication devices, new energy transportation vehicles, and aerospace, due to their high energy density, stability, durability, and affordability [1,2]. Over time, the battery's internal resistance increases, causing a decline in its performance and ultimately compromising the electrical equipment's safety [3,4].

Currently, remaining useful life (RUL) prediction methods for lithium-ion batteries are categorized into two forms: model-driven methods and data-driven methods [5,6]. Model-driven methods establish a model based on the battery's electrochemical mechanism and degradation process to predict its RUL [7]. Model-based methods are commonly employed to develop battery life degradation models that are rooted in electrochemical mechanisms, enabling more accurate representation of battery's electrochemical characteristics. Nevertheless, the utilization of these methods is often restricted due to the demand for specialized expertise and battery design parameters, impeding their broader applicability [8].

With data-driven methods, lithium-ion battery monitoring data can be directly analyzed, identifying battery performance change patterns and predicting its RUL [9,10].

Ren et al. [11] proposed an RUL prediction method for lithium-ion batteries leveraging Auto-CNN-LSTM. Zhang et al. [12] proposed a deep learning-based method for predicting the RUL of lithium-ion batteries using LSTM and a recurrent neural network (RNN).

Lithium-ion batteries experience problems with capacity recovery during their degradation process. To address this concern, researchers have employed signal processing methodologies to preprocess the battery's capacity sequence data [13]. Li et al. [14] proposed an algorithm based on empirical mode decomposition (EMD) combined with Elman–LSTM for predicting the RUL of lithium-ion batteries. Meng et al. [15] proposed an algorithm based on complete ensemble empirical mode decomposition with adaptive noise (CEEMDAN) combined with an adaptive neuro-fuzzy inference system (ANFIS) for precise lithium-ion battery capacity prediction. Lyu et al. [16] proposed a scheme called the VPA model for predicting the RUL of lithium-ion batteries. Their model uses the variational mode decomposition algorithm (VMD) to obtain the trend signals and capacity regeneration signals. Particle filter and autoregressive moving average models are then used to predict the two signals separately, and their predicted results are fused to obtain the overall capacity degradation prediction.

Based on the aforementioned analysis, utilizing a prediction model that utilizes both signal processing algorithms and data-driven methods can effectively improve the prediction accuracy of RUL. By selecting appropriate signal processing algorithms and data-driven methods, it is possible to significantly enhance prediction performance. VMD [17] can effectively mitigate the problems associated with pattern aliasing and endpoint effects. Additionally, VMD has high decomposition efficiency and is highly resistant to noise. Therefore, in this paper we adopt VMD as the signal processing algorithm to process the capacity attenuation signal of lithium batteries [18–20]. Long- and short-term time-series networks (LSTNet) [21] have recently been widely adopted in diagnostic and predictive scenarios due to its excellent performance.

The effectiveness of VMD decomposition is mainly influenced by two factors, namely, the number of mode components ($K$) and the number of penalty factors ($\alpha$) [22]. Similarly, the performance of the LSTNet model is significantly affected by its parameter values [23]. Although the sparrow search algorithm (SSA) is straightforward to implement, it can be prone to becoming stuck in local optima [24]. Jia et al. [25] proposed the improved sparrow search algorithm (ISSA), which combines the elite opposition-based learning (EOBL) and Cauchy Gaussian Mutation strategies. The objective of ISSA is to enhance the diversity of sparrow populations and prevent them from becoming trapped in local optima. Qiao et al. [26] proposed an ISSA with firefly search disturbance by incorporating an iterative strategy from the Firefly algorithm to address the limitations of the original SSA.

In conclusion, this paper proposes a hybrid model for predicting the RUL of lithium-ion batteries by improving the VMD–LSTNet algorithm to accurately capture the phenomenon of rapid increases and decreases in battery capacity and to address problems with insufficient prediction accuracy. The paper's primary contributions are as follows:

(1) A novel method for predicting the RUL of lithium-ion batteries is proposed. First, the VMD algorithm is enhanced to decompose the measured battery capacity sequence into its trend components and capacity regeneration components. Additionally, trend components are forecast using the LSTNet–Attention model, whereas the LSTNet–Skip model is leveraged for predicting the capacity regeneration components. Lastly, the predicted results of each component are integrated to complete the battery RUL prediction process. The proposed approach addresses the challenge of insufficient accuracy in single models and inability to predict the complete trend of battery degradation.

(2) In order to overcome the limitation of SSA's susceptibility to local optima, we introduce an enhanced SSA algorithm that optimizes the respective positions of the initial population, producers, and scroungers within the traditional SSA framework.

(3) To enhance the decomposition effectiveness of VMD, we employ the minimum envelope entropy as the fitness function for ISSA. By optimizing the decomposition mode number $K$ and the penalty factor $\alpha$ of the VMD algorithm, the subsequent prediction

algorithms are able to capture the decomposed components more efficiently, thereby enhancing prediction accuracy.

(4) To address the issue of manual parameter adjustment in LSTNet, we employ ISSA to optimize its hyperparameters. Leveraging the distinct characteristics of the trend components and capacity recovery components, we employ the ISSA–LSTNet–Attention model and the ISSA–LSTNet–Skip model for prediction purposes.

These enhancements enable more accurate prediction of the RUL of lithium-ion batteries and offer more effective tools and methods for battery performance evaluation and maintenance. The validation and in-depth analysis of these innovative contributions will be conducted in subsequent experiments and related research.

## 2. Methods

### 2.1. Variational Mode Decomposition

There are fundamental differences in the decomposition principles between variational mode decomposition (VMD) and empirical mode decomposition (EMD). VMD is a non-recursive signal decomposition method that can specify the number of modes based on requirements [27]. The VMD algorithm involves constructing and solving variational problems. VMD has strong decomposition and anti-interference abilities, producing favorable results in non-stationary signal processing. Long-term use of lithium batteries results in the phenomenon of capacity recovery, requiring VMD application to mitigate its effect.

The first step in $f(t)$ signal processing is constructing a variational problem. The fundamental constraints for problem solving consist of:

$$
\begin{cases}
min_{(u,w)} \left\{ \sum_{k=1}^{K} \partial_t \left[ \left( \delta(t) + \frac{j}{\pi} \right) * u_k(t) \right] * e^{-jw_k t} \right\} ||_2^2 \\
\text{s.t.} \sum_{k=1}^{K} u_k(t) = f(t)
\end{cases}
\tag{1}
$$

where $\{u_k\}$ and $\{w_k\}$ represent the decomposed $K$ intrinsic mode functions (IMF) and their corresponding center frequencies, respectively. The $*$ symbol refers to a convolutional operation, $\partial_t$ represents taking the partial derivative of the time function $t$, $\partial_t$ represents the Dirac distribution function, and $f(t)$ is a signal requiring decomposition.

Incorporating the penalty factor $\alpha$ and Lagrange multiplier operator $\lambda$ while solving the variational problem in Equation (1) transforms the restrained variational problem into an unrestrained one.

$$
\begin{aligned}
&\Gamma(\{u_k\},\{w_k\},\lambda) = \\
&\alpha \left\{ \sum_{k=1}^{K} \partial_t \left[ \left( \delta(t) + \frac{j}{\pi} \right) * u_k(t) \right] * e^{-jw_k t} \right\} ||_2^2 + \\
&\left( f(t) - \sum_{k=1}^{K} u_t(t) \right) ||_2^2 - \left( \lambda(t), f(t) - \sum_{k=1}^{K} u_t(t) \right)
\end{aligned}
\tag{2}
$$

The values of $u_k^{N+1}$, $w_k^{N+1}$, and $\lambda^{N+1}$ are updated by Equation (2) using the alternating direction multiplier algorithm. The algorithm is iterated until the convergence condition is met.

$$
\widehat{u_k^{n+1}}(w) = \frac{\hat{f}(w) - \sum_{i \neq k} \widehat{u_i}(w) + \frac{\lambda(w)}{2}}{1 + 2\alpha(w - w_k)^2}
\tag{3}
$$

$$
w_k^{N+1} = \frac{\int_0^\infty w |\widehat{u_k}(w)|^2 dw}{\int_0^\infty |\widehat{u_k}(w)|^2 dw}
\tag{4}
$$

where $\widehat{u_k^{n+1}}(w)$ represents the Wiener filtering of the residual component, $w_k^{N+1}$ represents the center of gravity of the current mode component power spectrum, $\widehat{u_k}(w)$ denotes the Fourier transform, and $u_k(t)$ denotes the real part.

### 2.2. Sparrow Search Algorithm

The Sparrow Search Algorithm (SSA) is an optimization algorithm proposed based on the behavior of sparrows [28]. In the SSA, the sparrow population is categorized into producers, scroungers, and scouts. The formula used to update the producers' location is

$$X_{i,j}^{t+1} = \begin{cases} X_{i,j}^t e^{-i/(\alpha i_{\text{itcrmax}})}, & R_2 < S_T \\ X_{i,j}^t + QL, & R_2 \geq S_T \end{cases} \tag{5}$$

where $t$ denotes the current number of iterations, $X_{i,j}^t$ denotes the position of the $i$-th sparrow in the $j$-th dimension at the $t$-th iteration, $\alpha$ is a randomly generated number between 0 and 1, $i_{iter\max}$ denotes the maximum number of iterations, $R_2$ represents a predetermined warning threshold, and $S_T$ denotes the predetermined security threshold. If $R_2 < S_T$, the population is considered safe, whereas, if $R_2 \geq S_T$ the population is considered unsafe and needs to shift to a secure location. $Q$ is a random variable that follows a normal distribution.

The formula used to update the scroungers' location is

$$x_{i,j}^{t+1} = \begin{cases} Q \cdot \exp\left(\frac{x_{\text{worst}}^t - x_{i,j}^t}{i^2}\right), & i > \frac{N}{2} \\ X_P^{t+1} + \left| x_{i,j}^t - X_P^{t+1} \right| \cdot A^+ \cdot L, & i \leq \frac{N}{2} \end{cases} \tag{6}$$

where $X_{\text{worst}}^t$ represents the minimum position of the sparrow in the $i$-th iteration, $X_P^{t+1}$ represents the optimal position for a producer in the $(t+1)$-th iteration, and $A$ is a $1 \times d$ matrix where each element is randomly assigned as either 1 or $-1$.

The formula used to update the scouts' location is

$$X_{i,j}^{t+1} = \begin{cases} X_{\text{best}}^t + \beta \left| X_{i,j}^t - X_{\text{best}}^t \right|, & f_i > f_g \\ X_{i,j}^t + K \left[ \frac{\left| X_{i,j}^t - X_{\text{worst}}^t \right|}{(f_i - f_w) + \gamma} \right], & f_i = f_g \end{cases} \tag{7}$$

where $X_{\text{best}}^t$ represents the optimal position of a sparrow in the $i$-th iteration, $\beta$ is a random number generated from a normal distribution with a mean of 0 and a standard deviation of 1, the fitness value of the current sparrow individual is denoted by $f_i$, the value $\gamma$ represents an infinitely small constant, and the values $f_w$ and $f_g$ represent the current worst and best fitness values, respectively.

### 2.3. Improved Sparrow Search Algorithm

Despite its strong optimization performance, SSA is prone to falling easily into local optima. To address this issue, Tent Chaotic Mapping (TCM) is employed in this article to generate the initial population [29]. The positions of producers and scroungers are then optimized via Levy flight strategy (LF) [30].

#### 2.3.1. Tent Chaotic Mapping

The initial population of the SSA plays a crucial role in determining the overall search performance. Concentration and lack of randomness in the initial population may result in suboptimal search results. Improvements to the sentence structure and vocabulary were made in order to align with academic conventions, and the sentences were restructured to increase clarity and concision. Spelling and grammar were improved when necessary. The use of TCM to generate the initial population results in improved randomness and enhances both the global and diversity aspects of the search process. The specific mapping expression for this process is provided below:

$$X_{i+1} = \begin{cases} kX_i, 0 \leq X_i \leq 0.5 \\ k(1 - X_i), 0.5 < X_i \leq 1 \end{cases} \tag{8}$$

where $X_i$ denotes the value of the variable in the $i$-th iteration, while $k$ represents the parameter of the function of mapping.

### 2.3.2. Levy Flight Strategy

The SSA search scheme generation is significantly influenced by the location of the producers, and adoption of LF can facilitate more rapid identification of the global optimal solution. The producers' position has a significant impact on the generation of the SSA search scheme, while LF enables efficient identification of global optimum solution, presenting myriad exploration prospects. Thus, by optimizing the producers' position using LF, the algorithm's capability for searching the global optimum solution can be improved. Employing LF to optimize the scrounger position can enable the algorithm to break away from local convergence and approach the global optimum solution progressively.

The formula used to calculate the Levy flight step length, denoted as $s$, is shown below:

$$\begin{cases} s = \mu / |\beta|^{1/\beta}, 0 < \beta \leq 2 \\ \mu\left(0, \sigma_\mu^2\right), v(0, 1) \\ \sigma_\mu = \left\{ \frac{\Gamma(1+\beta) \cdot \sin(\pi\beta/2)}{\Gamma[(1+\beta)/2] \cdot \beta \cdot 2^{(\beta-1)/2}} \right\}^{1/\beta} \\ \Gamma(1 + \beta) = \int_0^\infty t^\beta \cdot e^{-t} dt \end{cases} \tag{9}$$

The formula used for updating the producers' position by optimizing the LF is presented below:

$$X_{i,j}^{t+1} = \begin{cases} X_{i,j}^t e^{-i/(\alpha i_{\text{itcrmax}})} + s \cdot \left| X_{i,j}^t - X_{\text{best}}^t \right|, & R_2 < S_T \\ X_{i,j}^t + QL + s \cdot \left| X_{i,j}^t - X_{\text{best}}^t \right|, & R_2 \geq S_T \end{cases} \tag{10}$$

The formula used for updating the scroungers' position by optimizing the LF is presented below:

$$x_{i,j}^{t+1} = \begin{cases} Q \cdot \exp\left(\frac{x_{\text{worst}}^t - x_{i,j}^t}{i^2}\right), & i > \frac{N}{2} \\ X_P^{t+1} + s \cdot \left| x_{i,j}^t - X_P^{t+1} \right| \cdot A^+ \cdot L, & i \leq \frac{N}{2} \end{cases} \tag{11}$$

### 2.4. LSTNet

The LSTNet model consists of both linear and nonlinear components [31]. The nonlinear component consists of the convolution module, the recurrent module, and the recurrent skip module; alternatively, an attention mechanism can be employed. The autoregressive (AR) model constitutes the linear component.

In this paper, the model that incorporates an attention mechanism in the nonlinear part is referred to as the LSTNet–Attention model, whereas the one utilizing the recurrent skip module is referred to as the LSTNet–Skip model. Figure 1 illustrates the network architecture of the LSTNet model.

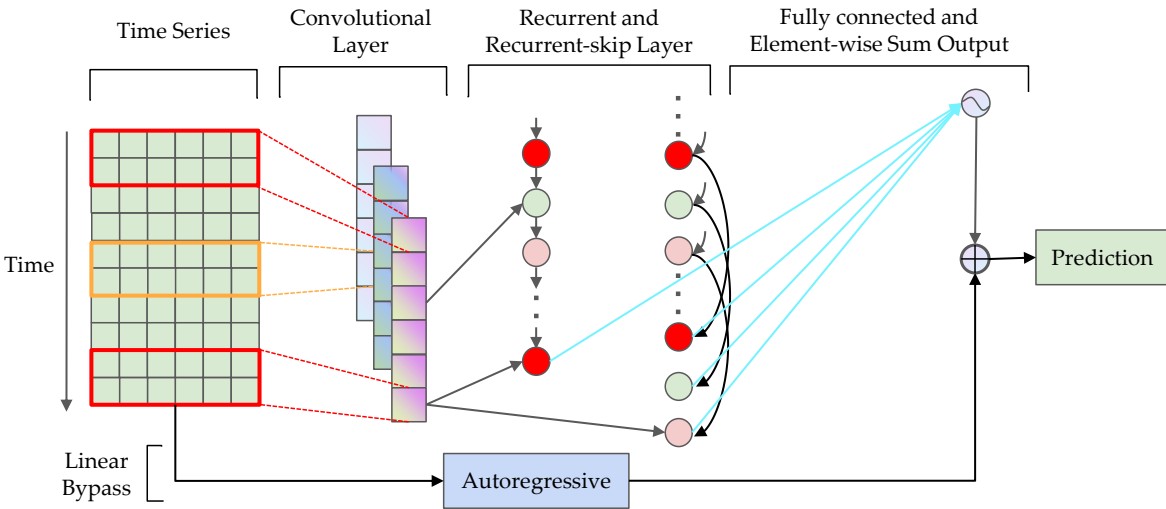

**Figure 1.** Internal structure of LSTNet.

### 2.4.1. Convolutional Module

The LSTNet convolutional module consists of a convolutional neural network that captures local information. The convolutional layer is comprised of several filters, with the formula for the *k*-th filter being:

$$h_k = \text{RELU}(W * X + b) \tag{12}$$

where the convolution operation is denoted by $*$. The output vector produced by this operation is $h_k$. The activation function employed is $RELU(x) = max(0, x)$ with an offset term $b$, and the input and output feature vectors are denoted as $X$ and $Y$, respectively.

### 2.4.2. Recurrent Module

Long Short-Term Memory Neural Networks (LSTM) represent an effective solution to the problems of Recurrent Neural Networks (RNN), specifically, gradient explosion and gradient disappearance. The core concepts of the LSTM neural network are Cell State and Gate, which function as filters for historical information and provide greater adaptability. The calculation process for LSTM is as follows:

$$f_t = \sigma\left(W_f h_{t-1} + U_f x_t + b_f\right) \tag{13}$$

$$i_t = \sigma(W_i h_{t-1} + U_i x_t + b_i) \tag{14}$$

$$\tilde{c}_t = RELU(W_c h_{t-1} + U_c x_t + b_c) \tag{15}$$

$$o_t = \sigma(W_o h_{t-1} + U_o x_t + b_o) \tag{16}$$

$$c_t = f_t \otimes c_{t-1} + i_t \otimes c_t \tag{17}$$

$$h_t = o_t \otimes RELU(c_t) \tag{18}$$

where $i_t$ represents the value of the input gate, $f_t$ represents the value of the forget gate, $o_t$ represents the value of the output gate, $c_t$ represents the value of the memory cell, and $b_i$, $b_f$, $b_o$, and $b_c$ represent their corresponding biases. The weight matrices $W$ and $U$ are utilized in the calculation process; $\sigma$ is a sigmoid function, $RELU$ is an activation function, and $\otimes$ represents an elementwise product.

2.4.3. Recurrent Skip Module

LSTM can capture dependencies in past information. However, as the length of the time series increases, issues arises when the gradient disappears, making it difficult for the LSTM to capture long-term patterns in the time series. To overcome this issue, a recurrent-skip module can be added.

The update process of the recurrent-skip module is expressed as follows:

$$r_t = \sigma\big(x_t\, W_{xr} + h_{t-p}\, W_{hr} + b_r\big) \tag{19}$$

$$u_t = \sigma\big(x_t\, W_{xu} + h_{t-p}\, W_{hu} + b_u\big) \tag{20}$$

$$o_t = \sigma\big(x_t\, W_{xo} + h_{t-p}\, W_{ho} + b_o\big) \tag{21}$$

$$\tilde{c}_t = \text{RELU}\big(x_t\, W_{xc} + h_{t-p}\, W_{hc} + b_c\big) \tag{22}$$

$$c_t = u_t \otimes c_{t-p} + r_t \otimes \tilde{c}_t \tag{23}$$

$$h_t = o_t \otimes \text{RELU}(c_t) \tag{24}$$

The input to the recurrent skip module equals the output of the convolution module, where $p$ represents the number of hidden layer states skipped.

Here, $h_t^R$ represents the output from the recurrent module at time $t$ and $\big\{h_{t-p+1}^S, h_{t-p+2}^S, \cdots, h_t^S\big\}$ represents the output from the recurrent-skip module at times $t - p + 1$ through $t$. Then, the fully connected layer is used to fuse the outputs from the recurrent module and recurrent skip module as the prediction results for the nonlinear section; the predicted results are as follows:

$$h_t^D = W^R h_t^R + \sum_{i=0}^{p-1} W_i^S h_{t-i}^S + b \tag{25}$$

where $h_t^D$ represents the predicted result of the nonlinear section for time $t$.

2.4.4. Attention Mechanism Module

Not all time series exhibit seasonal characteristics, and the recurrent skip module requires presetting hyperparameters. On the other hand, by utilizing an attention mechanism in place of the recurrent skip module, it becomes possible to extract the features that require focus in a time series. The weight of attention at time $t$ can be represented as follows:

$$\alpha_t = AttnScore\big(H_t^R, h_{t-1}^R\big) \tag{26}$$

where $H_t^R = \big[h_{t-q}^R, \ldots, h_{t-1}^R\big]$ represents the stacking matrix of the hidden state $h_{t-1}^R$ of the LSTM neural network and $AttnScore$ is the attention mechanism function.

The output of the attention mechanism module results from the concatenation of a weighted vector $\mathbf{c}_t = H_t \alpha_t$, a window hidden layer state, and a linear projection. The specific formula is

$$h_t^D = W\big[c_t; h_{t-1}^R\big] + b \tag{27}$$

2.4.5. Autoregressive Module

While the convolutional module and the recurrent skip module both possess nonlinear characteristics, their sensitivity to input data is limited. This shortcoming can reduce

prediction accuracy. Thus, the LSTNet model addresses this issue by adding an autoregressive (AR) model. The formula for the AR module is

$$h_{t,i}^L = \sum_{k=0}^{q^{ar}-1} W_k^{ar} y_{t-k,i} + b^{ar} \tag{28}$$

where the prediction result of the AR model is represented by $h_t^L$, the coefficients of the AR model are $W^{ar}$ and $b^{ar}$, and $q^{ar}$ denotes the size of the input window.

To obtain the final prediction result of LSTNet, both the output of the neural network part and the result of the AR model are overlaid:

$$\hat{Y}_t = h_t^D + h_t^L \tag{29}$$

with final prediction result at time *t* represented by $\hat{Y}_t$ .

## 3. Construction of ISSA–VMD–LSTNet Model

### 3.1. Experimental Data

The performance of the proposed algorithm was validated using two lithium-ion battery datasets with distinct electrode materials and discharge environments.

The experimental hardware setup included an AMD 5600X processor, 16 GB of RAM, NVIDIA GTX 1070, Windows 10 operating system, PyCharm 2021 IDE, Python 3.7 programming language, and Keras 2.9.0 library.

#### 3.1.1. Database 1

The first experimental dataset used in this study was obtained from the Center for Advanced Life Cycle Engineering (CALCE) [32]. Specifically, this study examined CS2_35, CS2_36, CS2_37, and CS2_38, for which the capacity decay curves and the number of cycles during discharge are presented in Figure 2. Table 1 lists the detailed specifications of the selected lithium-ion batteries from CALCE.

**Table 1.** Detailed specifications of the selected lithium-ion batteries from CALCE.

| Battery | Rated Capacity/Ah | Constant Charging Current/A | Charging Cut-Off Voltage/V | Discharging Current/A | Discharging Cut-Off Voltage/V | Failure Threshold/Ah |
|---|---|---|---|---|---|---|
| CS2_35 | 1.1 | 0.55 | 4.2 | 0.55 | 2.7 | 0.77 |
| CS2_36 | 1.1 | 0.55 | 4.2 | 0.55 | 2.7 | 0.77 |
| CS2_37 | 1.1 | 0.55 | 4.2 | 0.55 | 2.7 | 0.77 |
| CS2_38 | 1.1 | 0.55 | 4.2 | 0.55 | 2.7 | 0.77 |

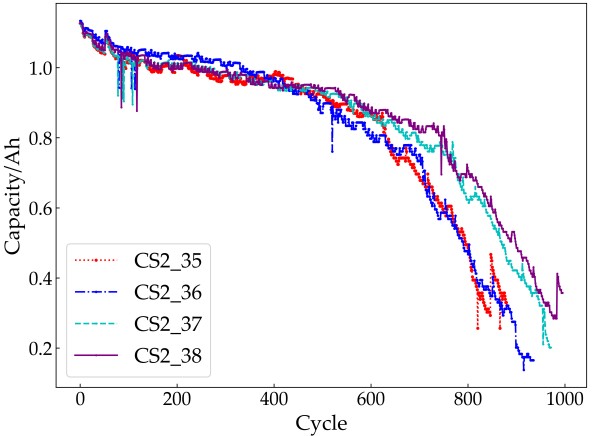

**Figure 2.** CALCE battery capacity decay curves with the number of cycles during discharge.

### 3.1.2. Database 2

The second experimental dataset used was the NASA lithium battery dataset, which includes the batteries B0005, B0006, B0007, and B0018 as the selected research objects [33]. Table 2 lists the detailed parameters of the selected NASA lithium battery dataset. It should be noted that the failure threshold of the B0007 battery is set to 1.45 Ah. Concerning the attenuation curve of capacity with the number of cycles during battery discharge, it is shown in Figure 3.

**Table 2.** Detailed specifications of the selected lithium-ion batteries from NASA.

| Battery | Rated Capacity/Ah | Constant Charging Current/A | Charging Cut-Off Voltage/V | Discharging Current/A | Discharging Cut-Off Voltage/V | Failure Threshold/Ah |
|---|---|---|---|---|---|---|
| B0005 | 2 | 1.5 | 4.2 | 2 | 2.7 | 1.4 |
| B0006 | 2 | 1.5 | 4.2 | 2 | 2.5 | 1.4 |
| B0007 | 2 | 1.5 | 4.2 | 2 | 2.2 | 1.45 |
| B0018 | 2 | 1.5 | 4.2 | 2 | 2.5 | 1.4 |

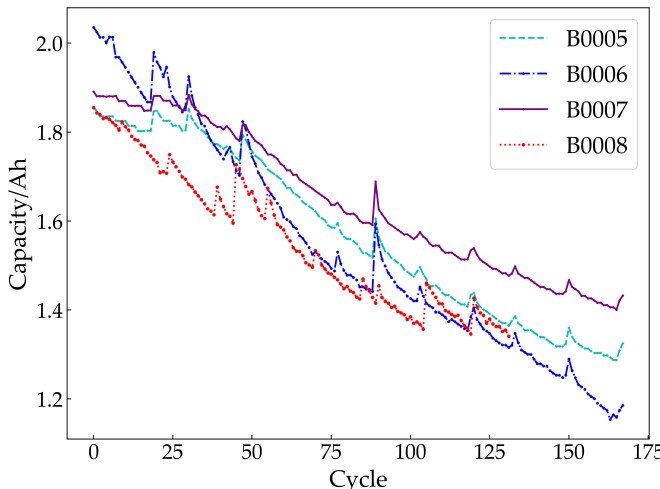

**Figure 3.** NASA battery capacity decay curves with the number of cycles during discharge.

### 3.2. Definition of RUL and Evaluation Criteria for Forecasting Methods

The RUL of a battery is defined as the number of remaining usable cycles from the predicted starting point to the end of the battery's life. When the actual capacity of the battery deteriorates to the failure threshold, the battery's life is considered to be over. $T$ is set as the starting cycle position, and $T_{\mathrm{EOL}}$ is the number of cycles at the end of the battery's remaining useful life in the actual state. The RUL of the battery is defined as

$$T_{\mathrm{RUL}} = T_{\mathrm{EOL}} - T \tag{30}$$

where the variable *EOL* represents the number of cycles that a battery can run at the conclusion of its remaining useful life period as estimated in advance.

The evaluation of the prediction model is based on four criteria: the root mean square error (RMSE), average absolute error (MAE), absolute correlation coefficient ($R^2$), and Nash–Sutcliffe efficiency index (NSE). These evaluation indicators are defined by the following formulas:

$$\mathrm{RMSE} = \sqrt{\frac{1}{n}\sum_{i=1}^{n}(x(i) - \hat{x}(i))^2} \tag{31}$$

$$\text{MAE} = \frac{1}{n} \sum_{i=1}^{n} |x(i) - \hat{x}(i)| \tag{32}$$

$$R^2 = 1 - \frac{\sum\limits_{i=1}^{n} (x_i - \hat{x}_i)^2}{\sum\limits_{i=1}^{n} (x_i - \bar{x})^2} \tag{33}$$

$$\text{NSE} = 1 - \frac{\sum\limits_{i=1}^{n} (x_i - \hat{x}_i)^2}{\sum\limits_{i=1}^{n} (x_i - \bar{x})^2} \tag{34}$$

where the number of cycles of the battery is represented by $n$ and the true and predicted values of the capacity sequence are represented by $x(i)$ and $\hat{x}(i)$, respectively.

These evaluation criteria are selected to assess the fitting and predictive accuracy of the model's prediction curve. Specifically, the smaller the values of MAE and RMSE, the closer to 1 the $R^2$ and NSE value will be, indicating that the model has a higher prediction accuracy [34].

*3.3. ISSA–VMD*

In this paper, the ISSA is used to optimize the number of mode components and penalty factors of VMD. The fitness function chosen for ISSA–VMD is the Minimum Envelope Entropy [35]. This function is represented by the following formula:

$$\begin{cases} E_P = -\sum\limits_{i=1}^{N} \varepsilon(i) \lg \varepsilon(i) \\ \varepsilon(i) = \frac{a(i)}{\sum\limits_{i=1}^{N} a(i)} \end{cases} \tag{35}$$

where $E_P$ represents the envelope entropy, $\varepsilon(i)$ represents the probability distribution sequence, $a(i)$ represents the envelope signal, and $N$ represents the number of sampling points.

The specific process of the ISSA–VMD method is as follows:

(1) Initialize the fundamental parameters of VMD and ISSA.

(2) The sparrow population is initialized using the TCM and the battery capacity sequence is subject to VMD decomposition, while the envelope entropy is utilized as the fitness function for conducting a global search.

(3) Update the positions of the producers, scroungers, and scouts using Equations (10), (11), and (7).

(4) Continuously execute steps 2 to 3 until the envelope entropy value reaches a minimum, then generate the current parameters $[k, \alpha]$.

(5) Use the optimal parameters to carry out VMD decomposition on the battery capacity sequence.

*3.4. ISSA–LSTNet*

In this paper, the ISSA method is utilized to optimize the key parameters of the convolution module, recurrent module, recurrent skip module, and AR module of the LSTNet–Attention model and LSTNet–Skip model:

$$\text{RMSE} = \sqrt{\frac{1}{n} \sum_{i=1}^{n} (x(i) - \hat{x}(i))^2} \tag{36}$$

where the number of cycles of the battery is represented by $n$ and the true and predicted values of the capacity sequence are represented by $x(i)$ and $\hat{x}(i)$, respectively.

The specific process of the ISSA–LSTNet method is as follows:

(1) Initialize the fundamental parameters for LSTNet–Attention, LSTNet–Skip, and ISSA.

(2) The sparrow population is initialized using the TCM, and the RMSE value serves as the fitness function to perform a global search.

(3) Update the positions of the producers, scroungers, and scouts using Equations (10), (11), and (7).

(4) Repeat steps 2 to 3 until the RMSE value reaches its minimum, then output the model parameters.

(5) Utilize the optimal parameters to forecast the RUL of the battery's capacity sequence.

### 3.5. ISSA–VMD–LSTNet Prediction Model

Figure 4 illustrates the architecture of the lithium-ion battery RUL prediction model proposed in this study. The RUL prediction method follows the steps outlined below.

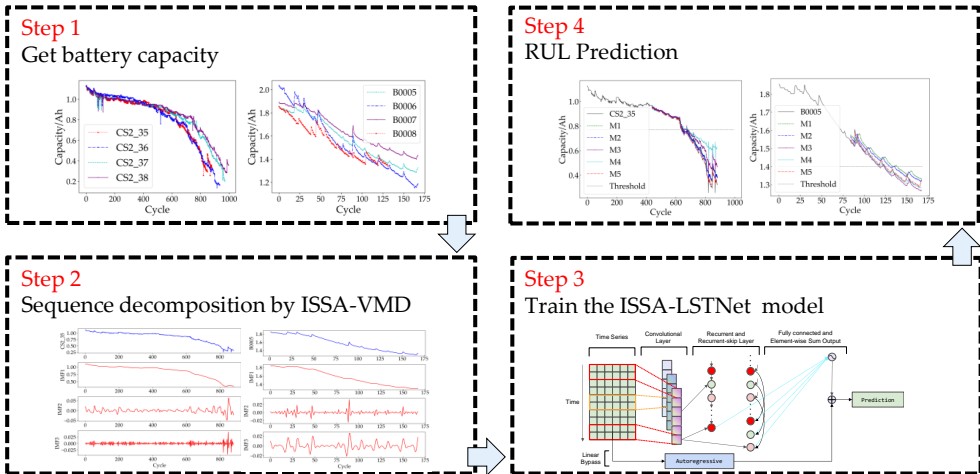

**Figure 4.** Framework overview of the proposed method.

(1) Collect the remaining discharge capacity data of the lithium-ion batteries.

(2) The capacity of the primary battery is decomposed into finite mode components using the ISSA–VMD algorithm. Next, a correlation analysis is conducted between the decomposed mode components and the degradation capacity sequence. The mode components with high correlation coefficients are considered trend components, while those with lower correlation coefficients are considered capacity recovery components.

(3) Train the trend and capacity recovery components separately using the ISSA–LSTNet–Attention model and ISSA–LSTNet–Skip model, respectively.

(4) Equation (37) enables seamless integration of the prediction results of the ISSA–LSTNet–Attention and ISSA–LSTNet–Skip models for accurate calculation of the RUL of lithium-ion batteries:

$$\hat{x}(i) = \sum_{j=1}^{n} IMF_j + IMF_{trend} \tag{37}$$

where $\hat{x}(i)$ represents the predicted value of the lithium battery capacity sequence, $IMF_{trend}$ represents the predicted value of the trend component, $n$ represents the total number of decomposed mode components, and $IMF_j$ represents the predicted value of the capacity recovery component for the $j$-th mode.

## 4. Results and Discussion

### 4.1. Decomposition of Lithium-Ion Battery Capacity Sequence by ISSA–VMD

ISSA-VMD is utilized to decompose the capacity attenuation curve based on the cycle number during the battery discharge process. The VMD parameters are optimized via ISSA $(k, \alpha)$, and their values are displayed in Table 3.

**Table 3.** Parameter settings for ISSA.

| Populations | Number of Iterations | Value Range of K | Value Range of $\alpha$ |
|---|---|---|---|
| 20 | 10 | [1, 10] | [1, 1000] |

#### 4.1.1. Database 1

Figure 5 displays the iterative process of optimizing VMD parameters using ISSA and SSA, with battery CS2_35 used as an illustration. The ISSA curve consistently remains at the bottom when compared to SSA, indicating fast convergence and effective optimization ability. The optimal parameter combination $(k, \alpha)$ is (3, 151). Figure 6 displays the original capacity sequence of the battery and the decomposed IMFs.

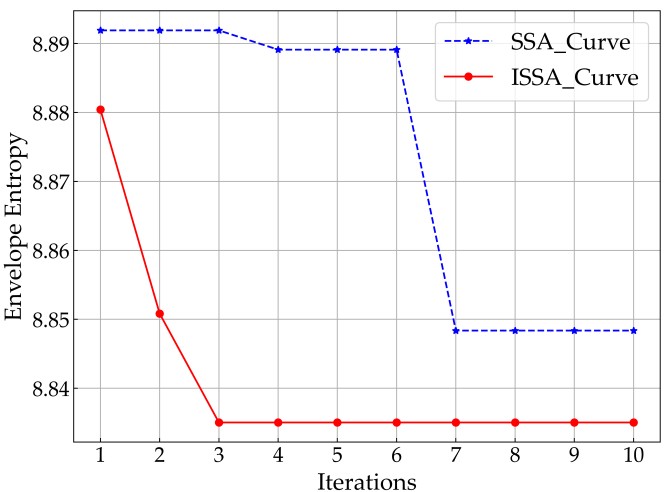

**Figure 5.** Change in fitness value for battery CS2_35.

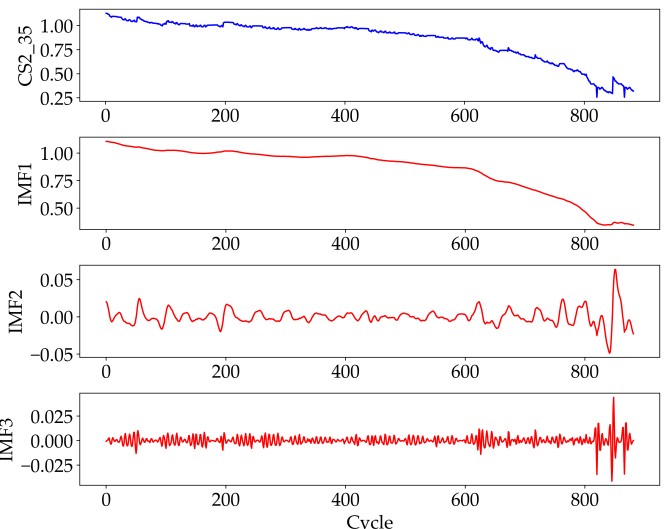

**Figure 6.** Capacity and decomposition results for battery CS2_35.

The final parameters resulting from the ISSA-optimized VMD process are presented in Table 4. In addition, we conducted a Pearson correlation analysis on the IMFs and the original capacity degradation sequence. The results are displayed in Table 5.

**Table 4.** Optimal parameters of ISSA–VMD.

| Battery | Number of Modes K | Penalty Factor $\alpha$ |
|---------|-------------------|-------------------------|
| CS2_35  | 3                 | 151                     |
| CS2_36  | 3                 | 247                     |
| CS2_37  | 3                 | 195                     |
| CS2_38  | 3                 | 183                     |

**Table 5.** Correlation coefficient between the original capacity sequence and the decomposed IMFs.

| Battery | IMF1     | IMF2     | IMF3     |
|---------|----------|----------|----------|
| CS2_35  | 0.997899 | 0.093756 | 0.038184 |
| CS2_36  | 0.999201 | 0.036894 | 0.026705 |
| CS2_37  | 0.998997 | 0.040341 | 0.032564 |
| CS2_38  | 0.998850 | 0.046070 | 0.032013 |

4.1.2. Database 2

Using battery B0005 as an example, the original capacity sequence of the battery was decomposed using ISSA–VMD. The shift of the fitness function in SSA–VMD and ISSA–VMD is illustrated in Figure 7, with the optimal parameter combination found to be $(k, \alpha) = (3, 1)$. The IMFs and the battery's original capacity sequence are displayed in Figure 8.

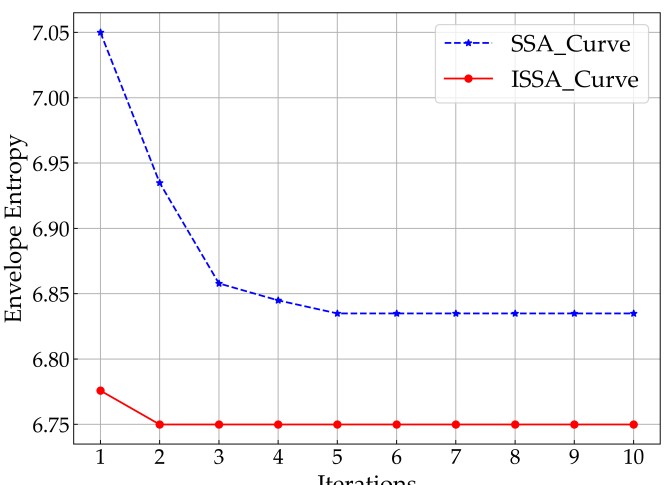

**Figure 7.** Change in fitness value for battery B0005.

Table 6 displays the definitive ISSA optimized VMD parameters. Furthermore, Table 7 exhibits the Pearson correlation analysis between the original capacity sequence and the decomposed IMFs. As a note, the IMF3 component was absent in batteries B0006 and B0018 during decomposition, and is represented by the symbol "-" in Table 7.

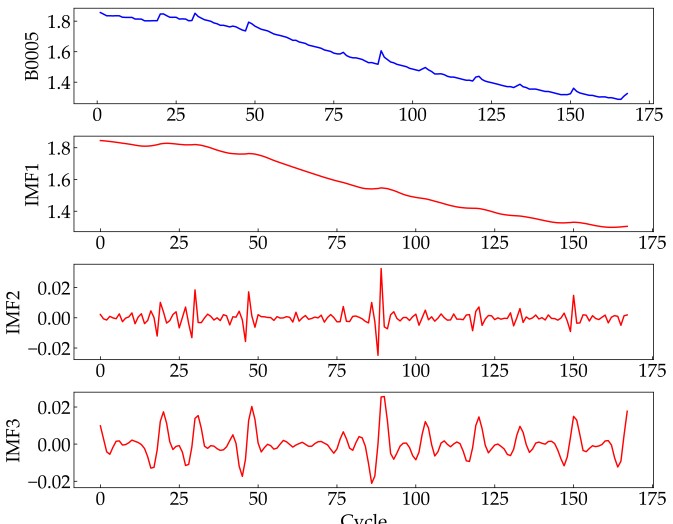

**Figure 8.** Capacity and decomposition results for battery B0005.

**Table 6.** Optimal parameters of ISSA–VMD.

| Battery | Number of Modes K | Penalty Factor $\alpha$ |
|---|---|---|
| B0005 | 3 | 1 |
| B0006 | 2 | 7 |
| B0007 | 3 | 60 |
| B0018 | 2 | 63 |

**Table 7.** Correlation coefficient between the original capacity sequence and the decomposed IMFs.

| Battery | IMF1 | IMF2 | IMF3 |
|---|---|---|---|
| B0005 | 0.998680 | 0.042675 | 0.071359 |
| B0006 | 0.998349 | 0.081128 | - |
| B0007 | 0.998278 | 0.077076 | 0.040673 |
| B0018 | 0.994283 | 0.129651 | - |

Tables 5 and 7 demonstrate that for IMF1, all correlation coefficients exceed 0.99, signifying a robust positive correlation between IMF1 and the original battery capacity sequence. Consequently, IMF1 can precisely capture the evolving trend of the original battery capacity sequence, making it an effective trend component that reflects the degradation of lithium batteries. However, with the exception of IMF1, the correlation coefficients between the remaining mode components and the original battery capacity sequence are all below 0.13, indicating very weak relationships. Nevertheless, these components can serve as indicators of capacity rebound and fluctuation.

*4.2. RUL Prediction*

The ISSA algorithm was employed in this study to optimize the LSTNet model parameters. The ISSA–LSTNet–Attention model was applied to train and predict the trend components, while the ISSA–LSTNet–Skip model was applied to train and predict the capacity recovery components. Table 8 presents the ISSA parameters.

**Table 8.** Parameter settings for ISSA.

| Populations | Number of Iterations | Value Range of Kernel Size | Value Range of Hidden Dimension | Value Range of Skip-Length | Value Range of Regularization Coefficient |
|---|---|---|---|---|---|
| 20 | 30 | [1, 10] | [1, 200] | [1, 10] | [1, 4] |

Table 9 presents the parameters acquired from the ISSA-optimized LSTNet model.

**Table 9.** Optimal parameters of ISSA–LSTNet.

| Module | Parameter Type | Parameter Settings |
|---|---|---|
| Convolution Module | kernel size | 2 |
| | hidden dimension | 24 |
| Recurrent Module | hidden dimension | 94 |
| Recurrent-Skip Module | hidden dimension | 66 |
| | skip-length | 2 |
| Autoregressive Module | regularization coefficient | 3 |

The CS2_35 and B0005 batteries were used as examples. The first half of the data was chosen for training, while the second half was chosen for testing. The LSTNet–Attention model was utilized to train the trend component derived from ISSA–VMD decomposition. On the other hand, the fluctuation component that was obtained from ISSA–VMD decomposition was input to LSTNet–Skip model for training. Following completion of the training phase, the prediction results of the LSTNet–Attention model and LSTNet–Skip model were combined utilizing Equation (37). Figures 9 and 10 display the prediction outcomes for each individual model, while Figures 11 and 12 signify the final prediction results.

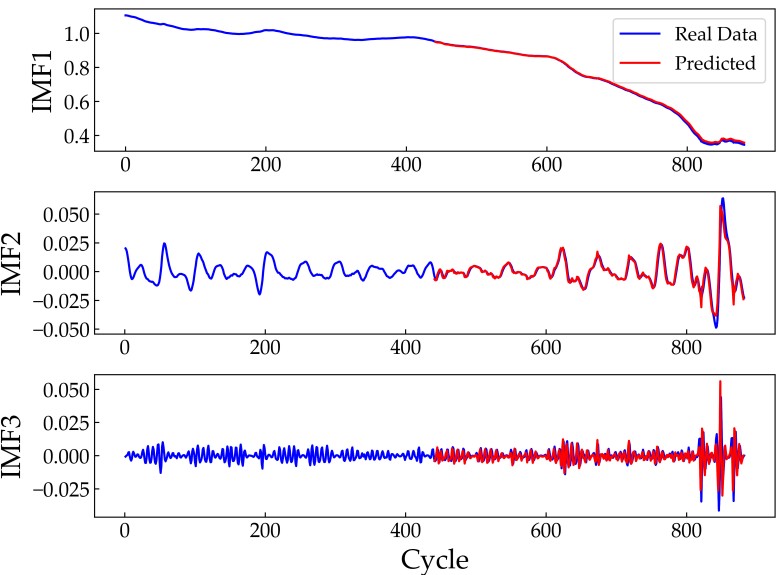

**Figure 9.** Prediction results for the trend component and capacity recovery component of the CS2_35 battery.

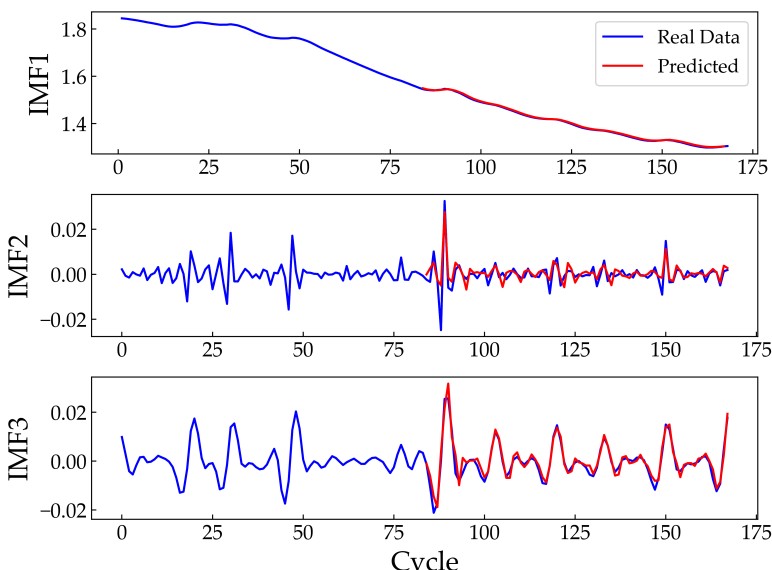

**Figure 10.** Prediction results for the trend component and capacity recovery component of the B0005 battery.

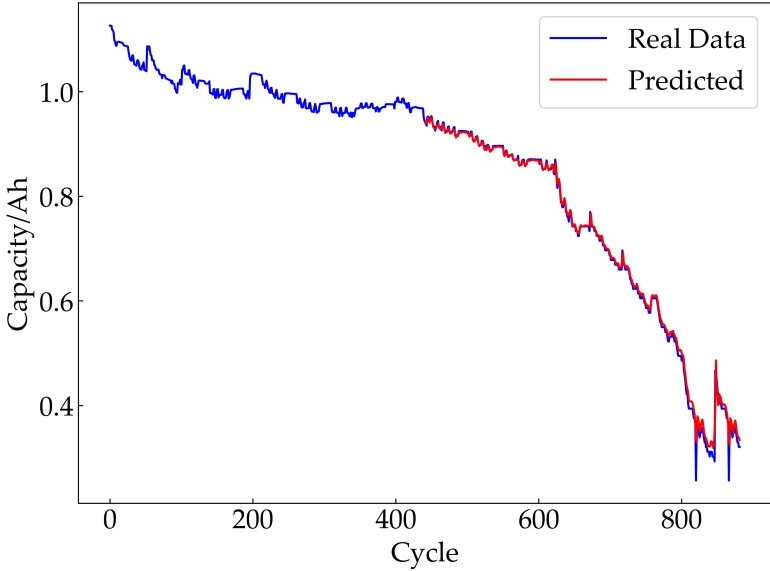

**Figure 11.** RUL prediction results for battery CS2_35.

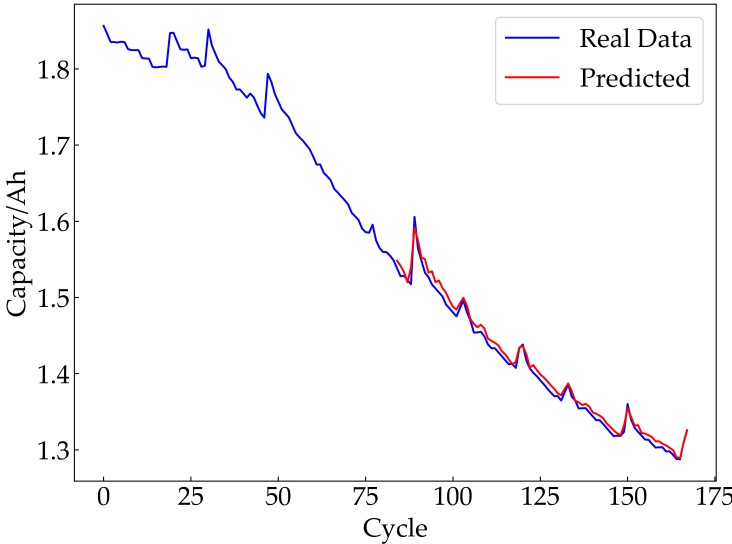

**Figure 12.** RUL prediction results for battery B0005.

### 4.3. Comparison and Analysis of Forecast Results

This article's proposed method (M5) was compared against four single prediction models (CNN, LSTM, LSTNet–Attention, and LSTNet–Skip) in order to verify its superiority. In this study, we refer to CNN, LSTM, LSTNet–Attention, and LSTNet–Skip as M1, M2, M3, and M4, respectively. The batteries were trained on the first 50% of the data and tested on the remaining 50%. We evaluated the prediction results of each model using MAE, RMSE, and $R^2$. The results are presented in Figures 13 and 14 and Tables 10 and 11.

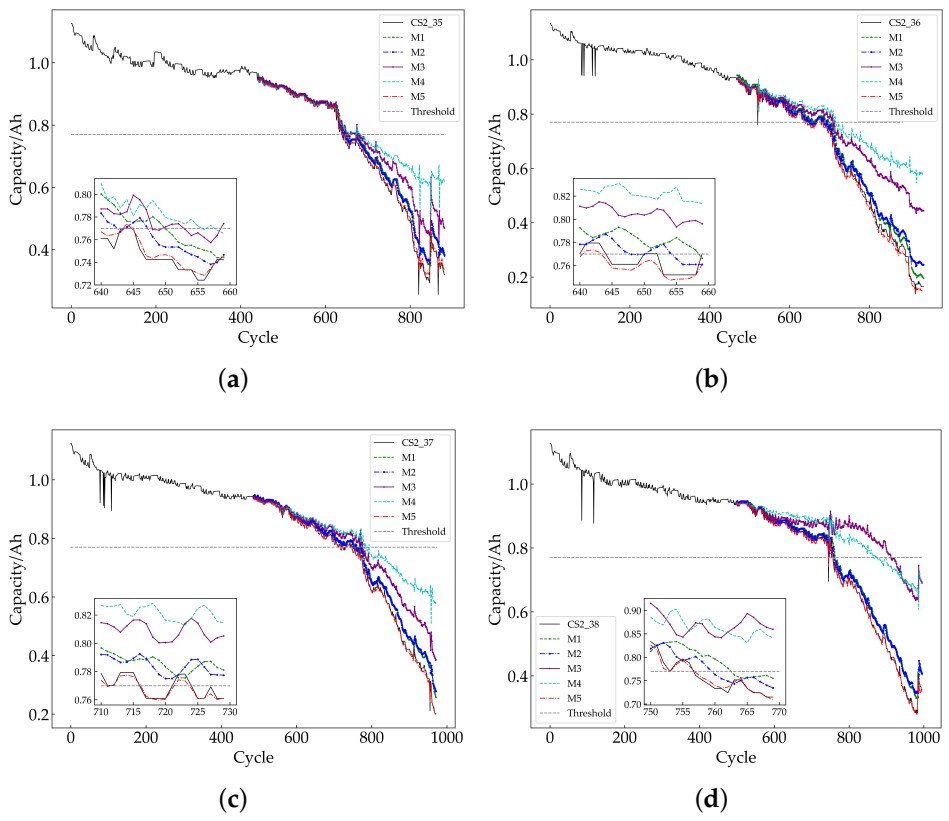

**Figure 13.** RUL prediction results for CALCE batteries. (**a**) CS2_35; (**b**) CS2_36; (**c**) CS2_37; (**d**) CS2_38.

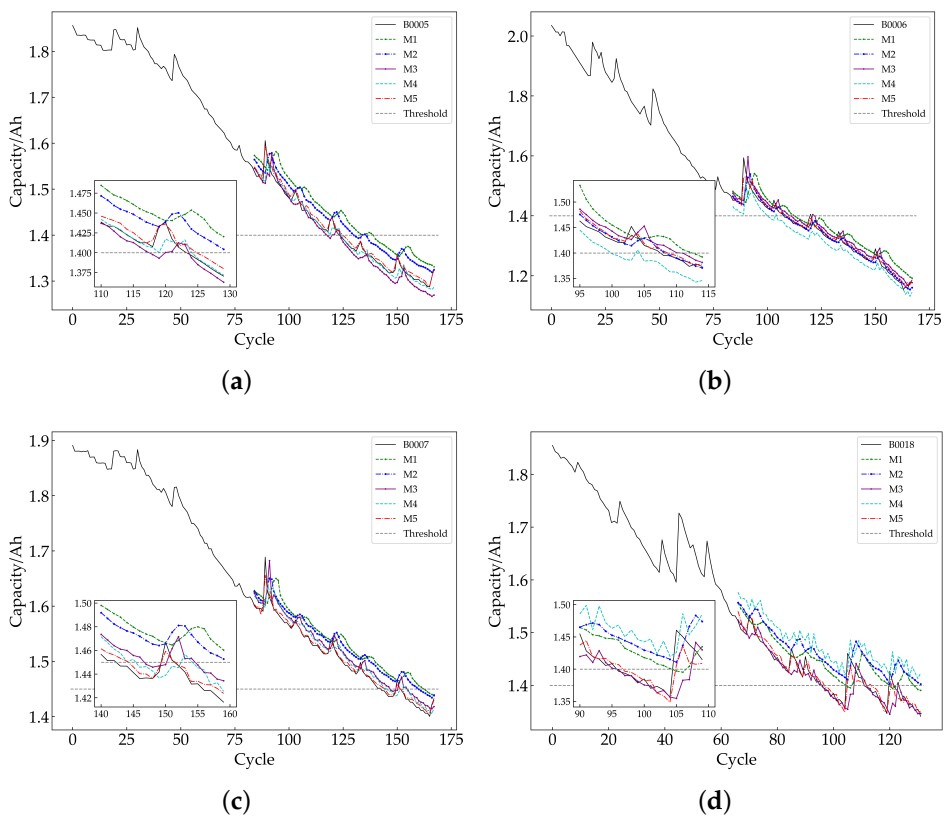

(**a**)  (**b**)

(**c**)  (**d**)

**Figure 14.** RUL prediction results for NASA batteries. (**a**) B0005; (**b**) B0006; (**c**) B0007; (**d**) B0018.

**Table 10.** Comparison of the evaluation indexes of the five methods on the CALCE dataset.

| Battery | Method | RMSE (%) | MAE (%) | $R^2$ (%) | NSE (%) |
|---|---|---|---|---|---|
| CS2_35 | M1 | 3.41 | 2.51 | 97.15 | 97.15 |
| | M2 | 2.96 | 2.1 | 97.85 | 97.85 |
| | M3 | 12.53 | 7.96 | 61.53 | 61.53 |
| | M4 | 7.27 | 5.05 | 87.04 | 87.04 |
| | M5 | 1.02 | 0.60 | 99.75 | 99.75 |
| CS2_36 | M1 | 3.32 | 2.79 | 97.96 | 97.96 |
| | M2 | 3.69 | 2.75 | 97.48 | 97.48 |
| | M3 | 18.76 | 13.94 | 35.02 | 35.02 |
| | M4 | 13.38 | 10.05 | 66.95 | 66.95 |
| | M5 | 1.31 | 1.03 | 99.68 | 99.68 |
| CS2_37 | M1 | 3.15 | 2.45 | 97.55 | 97.55 |
| | M2 | 3.16 | 2.41 | 97.54 | 97.54 |
| | M3 | 13.69 | 9.54 | 53.72 | 53.72 |
| | M4 | 8.06 | 5.97 | 83.96 | 83.96 |
| | M5 | 0.66 | 0.47 | 99.89 | 99.89 |
| CS2_38 | M1 | 3.09 | 2.5 | 97.57 | 97.57 |
| | M2 | 2.8 | 2.13 | 98 | 98 |
| | M3 | 17.61 | 13.18 | 21.21 | 21.21 |
| | M4 | 19.01 | 13.95 | 8.2 | 8.2 |
| | M5 | 0.67 | 0.40 | 99.89 | 99.89 |

**Table 11.** Comparison of the evaluation indexes of the five methods on the NASA dataset.

| Battery | Method | RMSE (%) | MAE (%) | $R^2$ (%) | NSE (%) |
|---------|--------|----------|---------|-----------|---------|
| B0005 | M1 | 4.35 | 4.13 | 70.62 | 70.62 |
| | M2 | 3.13 | 2.94 | 84.77 | 84.77 |
| | M3 | 1.54 | 0.90 | 96.33 | 96.33 |
| | M4 | 2.11 | 1.51 | 93.08 | 93.08 |
| | M5 | 0.82 | 0.71 | 98.95 | 98.95 |
| B0006 | M1 | 3.95 | 3.37 | 84.15 | 84.15 |
| | M2 | 2.30 | 1.21 | 94.61 | 94.61 |
| | M3 | 3.57 | 2.84 | 86.99 | 86.99 |
| | M4 | 2.65 | 2.04 | 92.84 | 92.84 |
| | M5 | 1.23 | 0.89 | 98.45 | 98.45 |
| B0007 | M1 | 3.53 | 3.27 | 70.21 | 70.21 |
| | M2 | 2.98 | 2.78 | 78.73 | 78.73 |
| | M3 | 1.69 | 1.33 | 93.17 | 93.17 |
| | M4 | 1.95 | 1.64 | 90.89 | 90.89 |
| | M5 | 0.58 | 0.41 | 99.17 | 99.17 |
| B0018 | M1 | 3.79 | 3.52 | 39.48 | 39.48 |
| | M2 | 4.79 | 4.58 | 3.39 | 3.39 |
| | M3 | 6.13 | 5.87 | −58.09 | −58.09 |
| | M4 | 2.43 | 1.41 | 75.23 | 75.23 |
| | M5 | 1.29 | 0.95 | 92.93 | 92.93 |

### 4.3.1. Database 1

From the data presented in Figure 13, it is evident that the CS2_35, CS2_36, CS2_37, and CS238 batteries all show a level of capacity recovery. The curve for the method proposed in this paper has a better fit with the actual capacity degradation curve of the battery than the curves of the other single models, as illustrated in Figure 13.

Table 10 shows that the proposed model outperforms the single LSTNet–Skip model in terms of the average RMSE, average MAE, and average $R^2$. The average RMSE, average MAE, and average $R^2$ of the proposed model are 0.915, 0.625, and 99.8025, respectively, compared to 15.6475, 11.155, and 42.87 for the single LSTNet–Skip model. This represents an improvement in accuracy of 14.733 and 10.53, respectively, for the average RMSE and average MAE, and an increase in accuracy of 56.9625 for the average $R^2$. These results provide strong evidence that the proposed method significantly outperforms other models.

### 4.3.2. Database 2

Figure 14 indicates that a single model is susceptible to interference because of the capacity rebound phenomenon observed in batteries B0005, B0006, B0007, and B0018. Moreover, as illustrated in Figure 14a–c, a single model yields low accuracy in predicting battery capacity. Additionally, the changes in the instability of battery B0018 and the limited data available make it particularly challenging for the prediction model to learn previous patterns. Overall, the predictive abilities of the single models with respect to battery capacity are inferior to the accuracy achieved by the method proposed in this article.

As per Table 11, the single LSTNet–Skip model has an average RMSE of 3.2325, whereas the proposed model has an average RMSE of 0.98. Therefore, the proposed model shows a decrease of 2.252 in average RMSE. Similarly, the average MAE of the single LSTNet–Skip model is 2.735, while the proposed model averages 0.74, signaling a reduction of 1.995 in average MAE. Additionally, the average $R^2$ of the single LSTNet–Skip model is 54.6, whereas the proposed model has an average $R^2$ of 97.375, indicating an increase of 42.775 on average $R^2$. These results demonstrate that the technique introduced in this paper can effectively enhance accuracy when predicting the RUL of lithium batteries.

To further illustrate the superior performance of our proposed method, we conducted a comparison between it and approaches previously reported in the literature. Tang et al. [36] proposed a hybrid model based on CEEMDAN–IGWO–BiLSTM to predict the RUL of batteries, while Hu et al. [37] proposed an RUL prediction method for lithium-ion batteries based on DEGWO–MSVR. Tables 12 and 13 compare the prediction results of the aforemen-

tioned two algorithms and the proposed hybrid method (unavailable data in references are denoted by "-").

**Table 12.** Comparison of our algorithm with other RUL prediction algorithms on the CALCE dataset.

| Battery | Method | RMSE (%) | MAE (%) | $R^2$ (%) | NSE (%) |
|---------|--------|----------|---------|-----------|---------|
| CS2_35 | CEEMDAN–IGWO–BiGRU | 2.58 | 1.35 | 98.96 | - |
| | DEGWO-MSVR | 2.37 | 1.97 | - | - |
| | Proposed method | 1.02 | 0.60 | 99.75 | 99.75 |
| CS2_36 | CEEMDAN–IGWO–BiGRU | 2.03 | 1.57 | 99.23 | - |
| | DEGWO-MSVR | 2.9 | 2.16 | - | - |
| | Proposed method | 1.31 | 1.03 | 99.68 | 99.68 |
| CS2_37 | CEEMDAN–IGWO–BiGRU | 1.92 | 1.36 | 99.06 | - |
| | DEGWO-MSVR | 2.58 | 1.91 | - | - |
| | Proposed method | 0.66 | 0.47 | 99.89 | 99.89 |
| CS2_38 | CEEMDAN–IGWO–BiGRU | 1.91 | 1.36 | 99.06 | - |
| | DEGWO-MSVR | 2.35 | 1.80 | - | - |
| | Proposed method | 0.67 | 0.40 | 99.89 | 99.89 |

**Table 13.** Comparison of our algorithm with other RUL prediction algorithms on the NASA dataset.

| Battery | Method | RMSE (%) | MAE (%) | $R^2$ (%) | NSE (%) |
|---------|--------|----------|---------|-----------|---------|
| B0005 | CEEMDAN–IGWO–BiGRU | 4.91 | 2.53 | 93.86 | - |
| | DEGWO-MSVR | 1.19 | 1.03 | - | - |
| | Proposed method | 0.82 | 0.71 | 98.95 | 98.95 |
| B0006 | CEEMDAN–IGWO–BiGRU | 4.90 | 1.99 | 91.87 | - |
| | DEGWO-MSVR | 2.24 | 2.03 | - | - |
| | Proposed method | 1.23 | 0.89 | 98.45 | 98.45 |
| B0007 | CEEMDAN–IGWO–BiGRU | 4.95 | 2.47 | - | - |
| | DEGWO-MSVR | 0.75 | 0.63 | - | - |
| | Proposed method | 0.58 | 0.41 | 99.17 | 99.17 |
| B0018 | CEEMDAN–IGWO–BiGRU | 5.49 | 3.62 | 72.29 | - |
| | DEGWO-MSVR | 0.69 | 0.59 | - | - |
| | Proposed method | 1.29 | 0.95 | 92.93 | 92.93 |

The CEEMDAN–IGWO–BiGRU model aligns with the objective of this article. Initially, the decomposition algorithm is employed to decompose the lithium battery capacity degradation sequence into a global degradation trend component and local capacity recovery component. Subsequently, an intelligent optimization algorithm is utilized to optimize the hyperparameters of the neural network. Finally, the prediction results of the various models are integrated. However, this method does not consider selection of an appropriate prediction model based on the characteristics of the trend components and capacity recovery components, which leads to limited generalization capability. Table 12 demonstrates that the proposed model achieves better performance compared to the CEEMDAN–IGWO–BiGRU model, with reductions of 1.195 and 0.785 in average RMSE and average MAE, respectively, and an increase of 0.725 in average $R^2$ and average NSE. Moreover, Table 13 reveals that the proposed model outperforms the CEEMDAN–IGWO–BiGRU model, with reductions of 4.0825 and 1.9125 in average RMSE and average MAE, respectively, and an increase of 9.55 in average R2 and average NSE. These results indicate that the proposed method significantly improves prediction accuracy compared to other models.

The DEGWO–MSVR model utilizes DEGWO to optimize Multi-Kernel Support Vector Regression (MSVR), which integrates multiple kernel functions to construct support vector regression (SVR) prediction models. However, this approach fails to address the issue of capacity rebound, leading to interference in its prediction accuracy. Table 12 illustrates that the model proposed in this paper achieves reductions of 1.635 and 1.335 in average RMSE and average MAE, respectively, compared to the DEGWO–MSVR model. Similarly, Table 13 shows reductions of 0.2375 and 0.33 in average RMSE and average MAE,

respectively, compared to the DEGWO–MSVR model. These results provide substantial evidence supporting the significantly improved prediction accuracy of our proposed method compared to other models.

According to Tables 12 and 13, our proposed algorithm surpasses the other two comparative algorithms in various measures. The findings demonstrate that the ISSA–VMD–LSTNet method can proficiently capture the capacity recovery phenomenon of lithium-ion batteries and precisely forecast their RUL.

## 5. Conclusions

This paper presents an RUL prediction model for lithium batteries named ISSA–VMD–LSTNet. The findings of this research are as follows:

(1) This article introduces the ISSA, a modification of the SSA, which generates an initial population using TCM and optimizes the positions of discoverers and followers through LF. This addresses the SSA's susceptibility to fall into local optima.

(2) The effectiveness of VMD decomposition is improved by adopting ISSA to optimize the number of modes and penalty factors of VMD. ISSA–VMD separates battery capacity data into a trend component and capacity recovery component, mitigating the adverse effects of the latter on model prediction.

(3) Optimizing the LSTNet model parameters with ISSA enhances the predictive performance of the model. The decomposed IMFs are predicted using LSTNet–Attention and LSTNet–Skip models, then their prediction results are integrated to eliminate the vulnerability of single model prediction accuracy to interference.

(4) The experimental results demonstrate the significant advantages of the ISSA–VMD–LSTNet model in predicting the RUL of lithium batteries, resulting in a notable enhancement in model accuracy. The proposed model was evaluated using two widely used lithium-ion battery datasets, yielding an RMSE below 2%, MAE below 1.5%, and $R^2$ and NSE exceeding 92%. These findings indicate that the proposed model exhibits superior prediction accuracy and performance compared to other models. Moreover, our research highlights the potential of the ISSA–VMD–LSTNet model to enhance the accuracy and stability of RUL prediction for lithium batteries.

(5) Finally, it is important to acknowledge the limitations of this study. In practical battery production, the RUL of lithium batteries is influenced by health factors, including current, voltage, and temperature. Hence, future experiments should encompass the consideration of multiple health factors and their impact on RUL prediction for lithium batteries. Subsequent research endeavors should focus on advancing and broadening the ISSA–VMD–LSTNet model in order to effectively tackle the challenges encountered in real-world battery operating conditions. Additionally, exploring additional model fusion strategies could further enhance prediction performance and stability.

**Author Contributions:** Conceptualization, X.Y.; methodology, X.Y.; software, X.Y.; validation, X.Y. and Y.L.; formal analysis, X.Y. and Y.L.; investigation, X.Y.; resources, X.Y.; data curation, X.Y. and Y.L.; writing—original draft preparation, X.Y.; writing—review and editing, H.W. and G.Z.; visualization, X.Y. and Y.L.; supervision, H.W. and G.Z.; project administration, H.W.; funding acquisition, H.W. All authors have read and agreed to the published version of the manuscript.

**Funding:** This work was supported in part by the National Natural Science Foundation of China (61863016).

**Data Availability Statement:** The data used in this paper are openly available. NASA dataset: https://www.nasa.gov/content/prognostics-center-of-excellence-data-set-repository (accessed on 8 May 2023); CALCE dataset: https://calce.umd.edu/data#CS2 (accessed on 8 May 2023).

**Acknowledgments:** We would like to thank the NASA Data Center and CALCE Center for providing the datasets used in this study.

**Conflicts of Interest:** The authors declare no conflict of interest.

## Abbreviations

The following abbreviations are used in this manuscript:

| | |
|---|---|
| RUL | Remaining Useful Life |
| SSA | Sparrow Search Algorithm |
| TCM | Tent Chaotic Mapping |
| LF | Levy Flight |
| ISSA | Improved Sparrow Search Algorithm |
| VMD | Variational Mode Decomposition |
| LSTNet | Long- and Short-Term Time-Series Network |
| RMSE | Root Mean Square Error |
| MAE | Mean Absolute Error |
| $R^2$ | Absolute Correlation Coefficient |
| NSE | Nash–Sutcliffe Efficiency |
| EMD | Empirical Mode Decomposition |
| CEEMDAN | Complete Ensemble Empirical Mode Decomposition with Adaptive Noise |
| IGOW | Improved Grey Wolf Optimizer |
| BiGRU | Bi-directional Gated Recurrent Unit |
| DE | Differential Evolution |
| MSVR | Multi-Kernel Support Vector Regression |

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
