# Peer review of "Remaining Useful Life Prediction for Lithium-Ion Batteries Based on Improved Mode Decomposition and Time Series"

_sustainability, doi:10.3390/su15129176_

Round 1

Reviewer 1 Report

In the original submission, the authors proposed a novel integrated technique, i.e., ISSA-VMD-LSTNet, for predicting the remaining useful life of lithium-ion batteries. The methodological part of the submitted work is strong; however, I listed some comments that need to be acknowledged by the authors before a possible publication.

Please re-write Lines 7-12.

The authors are suggested to provide more numerical results in the Abstract.

Line 73 – Additionally.

Please re-write Lines 79-80.

Experimental data and explanations regarding the data utilized should be presented before the Results section.

The authors are strongly recommended to include an additional performance metric, namely the coefficient of efficiency (or Nash-Sutcliffe efficiency index) as it better reflects the model fit compared to the coefficient of determination (R²).

How the information provided in Table 5 and Table 7 was used in this paper. Please enhance the explanations given in Lines 339-343 to provide more clarity to the interested researchers.

A discussion of the obtained results concerning the existing literature is missing.

Please highlight the major aim and contributions of the study in the Conclusion section. Also, limitations and implications for future attempts should be presented.

Author Response

please see the attachement

Reviewer 2 Report

Title of the peer-reviewed manuscript: “Remaining Useful Life Prediction for Lithium-Ion Batteries Based on Improved Mode Decomposition and Time Series”.

The manuscript consists of an Abstract, Keywords, an Introduction section, three main parts, a Conclusion section, list of References from 36 titles, 33 of which were published during the last 5 years. The manuscript contains 14 Figures and 13 Tables.

The goals and objectives of the study is to increase the service life of lithium-ion batteries by improving the accuracy of predicting the remaining useful life.

The relevance and practical significance of the presented study is determined by the massive use of lithium-ion batteries in various industrial and domestic applications. Despite the fact that many scientific articles have recently been devoted to the solution of the problem considered in this paper, the presented studies will be of interest to specialists.

The studies were carried out by the method of mathematical processing of the results of a direct natural experiment.

Questions and recommendations:

1. I would like to note the high practical significance of the research results presented in the paper. However, at the same time, I recommend that the authors more clearly formulate the scientific novelty. In its present form, the paper looks like applying known methods to a known object.

2. Did the authors use any specialized software? If so, then it should be indicated.

3. I suggest that the authors supplement the Conclusion section with plans for your further research.

In general, the manuscript is well organized and can be recommended for publication after minor revision.

Author Response

Response to Reviewer 2 Comments

Dear Reviewer,

Thank you for reviewing our manuscript entitled "Remaining Useful Life Prediction for Lithium-Ion Batteries Based on Improved Mode Decomposition and Time Series". We sincerely appreciate your valuable feedback and suggestions, as they have greatly contributed to enhancing the quality and impact of our research. We have thoroughly examined each of your comments and are pleased to inform you that we have successfully addressed the required revisions. In light of your feedback, we have implemented the following revisions:

Point 1:  I would like to note the high practical significance of the research results presented in the paper. However, at the same time, I recommend that the authors more clearly formulate the scientific novelty. In its present form, the paper looks like applying known methods to a known object.

Response 1: We have carefully considered your suggestion and made revisions to the manuscript to enhance the clarity and comprehensiveness of our explanation regarding the scientific novelty of our work. Specifically, we have incorporated a more explicit and detailed statement highlighting the unique contribution of our research at the end of the introduction section. We believe that these modifications effectively convey the distinctiveness of our study. Based on your valuable feedback, the revised content is as follows (Lines 81-114):

In conclusion, this paper proposes a hybrid model for predicting the RUL of lithium-ion batteries by improving the VMD-LSTNet algorithm to accurately capture the phenomenon of rapid increase and decrease in battery capacity and address problems with insufficient prediction accuracy. The paper’s primary contributions are as follows:

(1)A novel method for predicting the RUL of lithium-ion batteries is proposed. Firstly, the VMD algorithm was enhanced to decompose the measured battery capacity sequence into its trend components and capacity regeneration components. Additionally, trend components are forecasted using the LSTNet-Attention model, whereas the LSTNet-Skip model is leveraged for predicting the capacity regeneration components. Lastly, the predicted results of each component are integrated to complete the battery RUL prediction. The proposed approach addresses the challenge of insufficient accuracy in single models and the inability to predict the complete trend of battery degradation.

(2) In order to overcome the limitation of SSA's susceptibility to local optima, we introduce an enhanced SSA algorithm that optimizes the positions of the initial population, discoverers, and followers within the traditional SSA framework.

(3) To enhance the decomposition effectiveness of VMD, we employ the minimum envelope entropy as the fitness function for ISSA. By optimizing the decomposition mode number K and the penalty factor α of the VMD algorithm. The subsequent prediction algorithms capture the decomposed components more efficiently, thereby enhancing prediction accuracy.

(4) To address the issue of manual parameter adjustment in LSTNet, we employ ISSA to optimize its hyperparameters. Leveraging the distinct characteristics of trend components and capacity recovery components, we employ the ISSA-LSTNet-Attention model and the ISSA-LSTNet-Skip model for prediction purposes.

These enhancements enable more accurate prediction of the RUL of lithium-ion batteries and offer more effective tools and methods for battery performance evaluation and maintenance. The validation and in-depth analysis of these innovative contributions will be conducted in subsequent experiments and result discussions.

Point 2: Did the authors use any specialized software? If so, then it should be indicated.

Response 2: We apologize for the oversight in not mentioning the specific professional software utilized in our research. In the revised manuscript, we have addressed this by incorporating relevant details in Section 3.1, titled "Experimental Data," where we provide a comprehensive description of the software employed for analysis purposes. This addition serves to enhance reader comprehension regarding the tools and techniques employed in our study. Based on your valuable feedback, the revised content is as follows (Lines 258-160):

The experimental hardware setup included an AMD 5600X processor, 16 GB of RAM, NVIDIA GTX 1070, Windows 10 operating system, PyCharm 2021 IDE, Python 3.7 programming language, and Keras 2.9.0 library.

Point 3: I suggest that the authors supplement the Conclusion section with plans for your further research.

Response 3: We appreciate your valuable suggestion to include future research plans in the conclusion section. In response, we have incorporated a dedicated paragraph that outlines our intended future direction. This addition aims to provide readers with valuable insights into potential areas for further investigation and development, building upon the findings and impacts of our current research. Based on your insightful feedback, the revised content is as follows (Lines 483-519):

This paper presents a RUL prediction model for lithium batteries named ISSA-VMD- 409 LSTNet. The findings of this research are as follows:

(1) This article introduces the ISSA, which generates an initial population using TCM and optimizes the positions of discoverers and followers through LF. This addresses the drawback of the SSA's susceptibility to falling into local optima.

(2) The effectiveness of VMD decomposition improves by adopting ISSA to optimize the number of modes and penalty factors of VMD. ISSA-VMD separates battery capacity data into trend components and capacity recovery components to mitigate the adverse effects of the latter on model prediction.

(3) Optimizing LSTNet model parameters with ISSA enhanced the predictive performance of the model. The decomposed IMFs are predicted using LSTNet-Attention and LSTNet-Skip models, and their prediction results are integrated to eliminate the vulnerability of single model prediction accuracy to interference.

(4) The experimental results demonstrate the significant advantages of the ISSA-VMD-LSTNet model in predicting the RUL of lithium batteries, resulting in a notable enhancement in model accuracy. The proposed model was evaluated using two widely used lithium-ion battery datasets, yielding RMSE below 2%, MAE below 1.5%, and R2 and NSE values exceeding 92%. These findings indicate that the model exhibits superior prediction accuracy and performance compared to other models. Moreover, our research highlights the potential of the ISSA-VMD-LSTNet model in enhancing the accuracy and stability of RUL prediction for lithium batteries.

(5) However, it is important to acknowledge the limitations of this study. In practical battery production, the RUL of lithium batteries is influenced by health factors, including current, voltage, and temperature. Hence, future experiments should encompass the consideration of multiple health factors and their impact on RUL prediction for lithium batteries. Subsequent research endeavors should focus on advancing and broadening the ISSA-VMD-LSTNet model to effectively tackle the challenges encountered in real-world battery operating conditions. Additionally, exploring additional model fusion strategies can enhance prediction performance and ensure stability.

We express our sincere gratitude for your positive evaluation of the manuscript's organization and publication potential, as well as your recognition of the practical significance of our research. Taking into account your valuable suggestion, we have diligently revised the manuscript to enhance its robustness and suitability for publication.

We firmly believe that these revisions have significantly elevated the overall quality and comprehensiveness of our manuscript. Once again, we extend our appreciation for your valuable feedback and the considerable time and effort you invested in reviewing our work. Your guidance has provided us with the opportunity to strengthen our research further. We are hopeful that our revised manuscript meets your expectations and eagerly await your further guidance.

Sincerely,

The Authors

Reviewer 3 Report

This paper discusses an interesting and realistic topic and developed a new model to predict the life of lithium-ion batteries. Overall, the paper is well-written and clear, paragraphs are organized. Some revisions can be helpful before publication, especially to clearly discuss the advantages and limitations compared with other models. Some questions and comments are as follows:

1.      For the introduction section, what are the pros and cons of the model-driven method compared with the data-driven method?

2.      For figure 2, the captions and notations are too small to be readable, please revise. Also, a schematic illustration should be brief and concise.

3.      For the conclusion section, the author should briefly discuss the model’s advances (compared with other models), contributions, and limitations. 

The English language is well-written, some minor revisions are needed.

Author Response

Response to Reviewer 3 Comments

Dear Reviewer,

Thank you for reviewing our manuscript entitled "Remaining Useful Life Prediction for Lithium-Ion Batteries Based on Improved Mode Decomposition and Time Series". We sincerely appreciate your valuable feedback and suggestions, as they have greatly contributed to enhancing the quality and impact of our research. We have thoroughly examined each of your comments and are pleased to inform you that we have successfully addressed the required revisions. In light of your feedback, we have implemented the following revisions:

Point 1: For the introduction section, what are the pros and cons of the model-driven method compared with the data-driven method?

Response 1: We appreciate your suggestion and have incorporated it into the revised manuscript by expanding the introduction section. In this expansion, we have discussed the advantages and disadvantages of model-driven methods, supported by relevant references. The modifications aim to provide a more comprehensive overview of the topic and enhance the reader's understanding. Based on your insightful feedback, the revised content is as follows (Lines 34-38):

Model-based methods are commonly employed to develop battery life degradation models that are rooted in electrochemical mechanisms, enabling a more accurate representation of battery's electrochemical characteristics. Nevertheless, the utilization of these methods is often restricted due to the demand for specialized expertise and battery design parameters, impeding their broader applicability.

Point 2: For figure 2, the captions and notations are too small to be readable, please revise. Also, a schematic illustration should be brief and concise.

Response 2: We sincerely apologize for any readability issues in the image title and annotations. After careful consideration of your suggestion, we made efforts to modify Figure 4 by adjusting the size of the subtitles and symbols. However, despite our multiple attempts, we regret to inform you that we have not achieved the expected improvements in terms of readability. Figure 4 comprises Figures 2, 3, 6, 8, 1, 13 (a), and 14 (a) from the paper. (Lines 329)

Point 3: For the conclusion section, the author should briefly discuss the model’s advances (compared with other models), contributions, and limitations. 

Response 3: Based on your feedback, we have enhanced the conclusion section by providing a concise overview of the advancements achieved by our proposed model in comparison to existing models. Furthermore, we have included a thorough assessment of the limitations associated with our model to ensure a comprehensive evaluation. The revised content is as follows (Lines 483-519):

This paper presents a RUL prediction model for lithium batteries named ISSA-VMD- 409 LSTNet. The findings of this research are as follows:

(1) This article introduces the ISSA, which generates an initial population using TCM and optimizes the positions of discoverers and followers through LF. This addresses the drawback of the SSA's susceptibility to falling into local optima.

(2) The effectiveness of VMD decomposition improves by adopting ISSA to optimize the number of modes and penalty factors of VMD. ISSA-VMD separates battery capacity data into trend components and capacity recovery components to mitigate the adverse effects of the latter on model prediction.

(3) Optimizing LSTNet model parameters with ISSA enhanced the predictive performance of the model. The decomposed IMFs are predicted using LSTNet-Attention and LSTNet-Skip models, and their prediction results are integrated to eliminate the vulnerability of single model prediction accuracy to interference.

(4) The experimental results demonstrate the significant advantages of the ISSA-VMD-LSTNet model in predicting the RUL of lithium batteries, resulting in a notable enhancement in model accuracy. The proposed model was evaluated using two widely used lithium-ion battery datasets, yielding RMSE below 2%, MAE below 1.5%, and R2 and NSE values exceeding 92%. These findings indicate that the model exhibits superior prediction accuracy and performance compared to other models. Moreover, our research highlights the potential of the ISSA-VMD-LSTNet model in enhancing the accuracy and stability of RUL prediction for lithium batteries.

(5) However, it is important to acknowledge the limitations of this study. In practical battery production, the RUL of lithium batteries is influenced by health factors, including current, voltage, and temperature. Hence, future experiments should encompass the consideration of multiple health factors and their impact on RUL prediction for lithium batteries. Subsequent research endeavors should focus on advancing and broadening the ISSA-VMD-LSTNet model to effectively tackle the challenges encountered in real-world battery operating conditions. Additionally, exploring additional model fusion strategies can enhance prediction performance and ensure stability.

Point 4: The English language is well-written, some minor revisions are needed. 

Response 4: We appreciate your positive evaluation of the manuscript's overall language quality. Following your suggestion, we have conducted a thorough review of the document and implemented minor modifications to enhance the clarity and consistency of the English language.

We express our sincere gratitude for your positive evaluation of the manuscript's organization and publication potential, as well as your recognition of the practical significance of our research. Taking into account your valuable suggestion, we have diligently revised the manuscript to enhance its robustness and suitability for publication.

We firmly believe that these revisions have significantly elevated the overall quality and comprehensiveness of our manuscript. Once again, we extend our appreciation for your valuable feedback and the considerable time and effort you invested in reviewing our work. Your guidance has provided us with the opportunity to strengthen our research further. We are hopeful that our revised manuscript meets your expectations and eagerly await your further guidance.

Sincerely,

The Authors

Reviewer 4 Report

This paper presents a RUL prediction model of LIBs. The method improves the prediction accuracy and reduces the comlexity of the series. Overall, the manuscript is well organized with the main conclusions and well supported by the results. It is recommended that the manuscript may be accepted in present form.

Author Response

Response to Reviewer 4 Comments

Dear Reviewer,

Thank you for reviewing our manuscript titled ''Remaining Useful Life Prediction for Lithium-Ion Batteries Based on Improved Mode Decomposition and Time Series''. We greatly appreciate your positive evaluation of the organization and the strong results presented in this paper, as well as your recognition of the improved prediction accuracy and reduced complexity achieved through the proposed method. Your feedback and suggestions have provided valuable encouragement.

We acknowledge and appreciate the thoroughness of your evaluation, and we are pleased to inform you that we have accepted your suggestion and are accepting the manuscript in its current form. We believe that the modifications made based on the feedback from previous reviewers, along with the emphasized contributions and progress highlighted in the paper, have significantly improved its quality and suitability for publication.

We would like to express our sincere gratitude for the time and effort you have dedicated to reviewing our work. Your insightful feedback has made a substantial contribution to the refinement of our manuscript. Once again, we extend our thanks for your valuable feedback and the dedication you have shown in reviewing our work.

Thank you for your continued support.

Sincerely,

The Authors

Round 2

Reviewer 1 Report

Thanks for addressing my comments. I do recommend the publication of this manuscript in its current form.